# Using Gamma Irradiation to Predict Sperm Competition Mechanism in *Bagrada hilaris* (Burmeister) (Hemiptera: Pentatomidae): Insights for a Future Management Strategy

**DOI:** 10.3390/insects14080681

**Published:** 2023-08-01

**Authors:** Chiara Elvira Mainardi, Chiara Peccerillo, Alessandra Paolini, Alessia Cemmi, René F. H. Sforza, Sergio Musmeci, Daniele Porretta, Massimo Cristofaro

**Affiliations:** 1Biotechnology and Biological Control Agency (BBCA) Onlus, Via Angelo Signorelli 105, 00123 Rome, Italy; ale.paolini1@gmail.com (A.P.); m.cristofaro55@gmail.com (M.C.); 2Department of Environmental Biology, University of Rome “La Sapienza”, 00185 Rome, Italy; daniele.porretta@uniroma1.it; 3Center of Agriculture, Food and Environment (C3A), University of Trento, 38010 San Michele all’Adige, Italy; chiara.peccerillo@unitn.it; 4FSN-FISS-SNI Laboratory, Italian National Agency for New Technologies, Energy and Sustainable Economic Development (ENEA), Via Anguillarese 301, 00123 Rome, Italy; 5European Biological Control Laboratory, (USDA-ARS-EBCL), United States Department of Agriculture, 810 Avenue du Campus Agropolis, 34980 Montferrier-sur-Lez, France; rsforza@ars-ebcl.org; 6SSPT-BIOAG-SOQUAS Laboratory, Italian National Agency for New Technologies, Energy and Sustainable Economic Development (ENEA), Via Anguillarese 301, 00123 Rome, Italy

**Keywords:** sterile insect technique, irradiation, biological control, insect pest, stink bug, sperm mixing, sperm precedence

## Abstract

**Simple Summary:**

The greatest challenge for modern agriculture is the management of the damage caused by insect pests, in alternative to chemical insecticides, which are the most widely used tools for controlling these pest species. Among the sustainable alternatives to chemical insecticides, the Sterile Insect Technique (SIT) can help to control the target population through the release of adults that have been rendered sterile via irradiation in the field. In order to evaluate the applicability of this technique, a great deal of knowledge about the biology of the target species is required. In this study, we investigated the mechanism of sperm competition in a polyandrous stink bug species, *Bagrada hilaris.* In view of the possible application of the SIT, it is important that irradiated males have the same potential to compete with wild non-irradiated ones, even at the post-copulation stage. The results obtained showed a mechanism of sperm mixing.

**Abstract:**

The stink bug, *Bagrada hilaris*, is a pest of mainly Brassicaceae crops. It is native to Africa and Asia and was recently reported as invasive in the southwestern part of the USA and in South America. There are no mitigation programs in place that do not involve pesticides. Therefore, much attention has recently been paid to the study of this species in order to identify sustainable and effective control strategies, such as the Sterile Insect Technique (SIT). In order to evaluate the suitability of the SIT on this pest, the mechanism of post-copulatory sperm competition was investigated. This is a polyandrous species, and it is thus important to understand whether irradiated males are able to compete with wild, e.g., non-irradiated, males for sperm competition after matings. Sperm competition was studied by sequentially mating a healthy virgin female first with a non-irradiated male, and then with a γ-irradiated (Co-60) one, and again in the opposite order. Males were irradiated at three different doses: 60, 80, and 100 Gy. The fecundity and fertility of the females, in the two orders of mating, were scored in order to perform an initial assessment of the success of sperm competition with a P2 index. Sperm from the non-irradiated male were utilized at the lowest irradiation doses (60 and 80 Gy), whereas the irradiated sperm were preferentially utilized at the highest dose (100 Gy). *Bagrada hilaris* exhibited high variability in P2 indexes, indicating a sperm-mixing mechanism.

## 1. Introduction

*Bagrada hilaris* (Burmeister) (Hemiptera: Pentatomidae), commonly known as the bagrada bug or painted bug, is an invasive insect pest. It is native to Africa, India, Pakistan, and parts of Asia [1]. It was first reported in California (USA) in 2008, and expanded to southwestern U.S. states, Hawaii, and the countries of Mexico, Chile, and Argentina [2,3,4,5]. Additionally, this pest has also been reported in southern Europe (Malta and, specifically, the Island of Pantelleria in Italy) [6]. Even if *B. hilaris* has a wide host range, it is known to feed primarily on crucifers (family Brassicaceae) [2]. In addition, it has been observed to damage other significant food crops, including corn, kale, arugula, sunflower, barley, oat, wheat, artichoke, beetroot, carrot, lettuce, pea, rooibos [7,8,9,10,11], and caper plants, in southern Europe [12]. Feeding damage is caused by the lacerate-and-flush method of feeding, in which cell tissue is damaged through the repetitive insertion of stylets between the epidermal layers of leaves [9], combined with the injection of saliva [13,14], thus causing the destruction of apical meristems, young leaves, and terminal growth points [15]. In Italy, this species is only present on the island of Pantelleria, where it caused extensive and significant damage on caper plants (*Capparis spinosa* L.) [16]. Regarding the biology of the species, *B. hilaris* has a peculiar oviposition behavior: the female lays barrel-shaped eggs [17,18], but, unlike many stink bugs, deposits eggs singly or in small groups (of up to 10) under the soil [1,9,19]. Polyandry has been highlighted [20], and few effective control methods are available for managing this invasive alien pest species [21]. Frequent applications of broad-spectrum insecticides [9,21] have shown some positive results in the USA, but they are costly and have a significant impact on the environment [13,22]. Although *B. hilaris* is known to be attacked by predators and parasitoids [10], there are no documented biological control methods in use. Different species of egg parasitoids (belonging to the genera *Trissolcus, Gryon*, and *Ooencyrtus*, in the Scelionidae family) were found on *B. hilaris* in their native range in Pakistan [23], and now their potential use as classical biocontrol agents is under evaluation [24,25,26]. Among them, the most promising candidate seems to be *Gryon aetherium* Talamas, which appears to be more host-specific [27]. This parasitoid is the only one that is capable of detecting bagrada eggs underground [28]. Another ecologically friendly strategy, the Sterile Insect Technique (SIT) has been successfully applied for the suppression of several pest species, as a basic component of area-wide integrated pest management [29]. The synergistic effects of classical biological control in combination with the SIT in an area-wide program could enhance the likelihood of successful suppression of *B. hilaris* [30]. The SIT is a species-specific pest control method based on mass breeding of the target pest, its sterilization through irradiation, and its release into the environment at regular intervals [31]. Reproductive infertility is typically induced by exposure to X-rays, electron beams, or γ-rays generated from Cobalt or Cesium sources [32,33]. However, the irradiation process can measurably reduce the quality of the insects. Considering the possible side effects of such treatment, a key point in the SIT evaluation is the modulation of the irradiation dose. The aim is to interfere only with the degree of sterility of the gonads [34], reducing the negative consequences that risk compromising the fitness of the mass-released sterile individuals [35].

The irradiation biology of Hemiptera requires further study. Currently, there are no field uses of the SIT in this insect order, mostly because of the risk of direct feeding damage to the host crop species from the release of sterile phytophagous hemipteran adults of an invasive pest species [36]. Nowadays, the available studies on the irradiation of hemipterans concern *Halyomorpha halys* Stål [37,38,39] and *Nezara viridula* L. [40,41,42]. In *H. halys*, egg sterility after the γ exposure of males has been found to be 54.3% at 16 Gy [38]; additionally, in *N. viridula*, the application of a dose of 16 Gy or higher can sterilize males by 99% [43]. Even if incomplete sterility is probably not sufficient for an eradication program, support of the use of the SIT in area-wide control programs is provided by the recording of cumulative mortality cases in F1 in previous studies on *H. halys* [37,38,39].

In a recent study, the irradiation of *B. hilaris* adults was considered a control technique for the first time [30]. The results showed that the minimum dose necessary to achieve complete sterility was 100 Gy, and 90% sterility was estimated when irradiating males at 64 Gy. Furthermore, although both sexes are sensitive to irradiation, the sterile eggs obtained by mating irradiated males with non-irradiated females can be used in a classical biological control context [30]. In this study, the effects of irradiation were evaluated on both newly emerged males and females: results clearly showed that irradiated males mating with non-irradiated females have a strong impact on fertility (measured in egg-hatching rate), but do not have any significant effect on fecundity (oviposition rate). Although these results confirm that irradiated males can perform a full mating act with fertile females [30], the occurrence of multiple female matings is a critical aspect that must be evaluated.

However, polyandry does not necessarily negate the basic principles of the technique [44,45,46]. Several aspects should be considered, such as sperm competition, the morphology of the female reproductive system, the presence of a cryptic choice to select the sperm of different males, and eventual differences in mating performances between irradiated vs. non-irradiated males [30,47,48]. In terms of sperm competition, sterile irradiated males must be able to produce sperm at exactly the same rate as wild non-irradiated males. The presence of spermathecae, allowing bagrada females to store and utilize sperm from different males, is a critical factor that could influence the success of eradication programs [49]. Indeed, if females mate multiple times with non-irradiated males and preferentially use the most recently received sperm for fertilization, the effect of mating between irradiated males and non-irradiated females may be lost [35]. The issue becomes the competitiveness of irradiated males, which is influenced by post-mating factors (the ability to induce mating refractoriness in females, sperm competition, and/or sperm precedence), depending on the species [35,50,51]. In insects, the last copulating male often gains access to a disproportionate number of subsequent fertilizations [52]. This cryptic sperm selection mechanism is called last male sperm precedence (LMSP) [53,54]. Sperm competition success is expressed as the proportion of offspring of the last mating male with the female; since, experimentally, there are two competing males, paternity is calculated by taking the second male as the reference, with an index defined as “P2” [55]. This index is between 0 and 1: when P2 is 0, all members of the offspring are attributed to the first male; when it is equal to 1, they are all attributed to the second male; it is equal to 0.5 when there is a promiscuous use of the sperm of the two males, with equal probability of both to fertilize the eggs [56].

This study aimed to investigate the mechanism underlying sperm competition in *B. hilaris*. Females mate with multiple males, and the question was whether the mating order could influence the paternity of the eggs. A virgin female was isolated, first with a non-irradiated male, and then with an irradiated male, and vice versa, and it was investigated which male’s sperm had access to fertilize the eggs. The fertility, fecundity, and percentage of eggs hatched were measured. This approach allowed us to first assess the

P2 values of this species. It was decided to test at three different doses (60, 80, and 100 Gy) because, in SIT programs, the most suitable dose is the tradeoff between effective sterility and male performance [57].

This research is part of a wider program aiming to assess the applicability of the SIT technique in synergy with classic biological control in order to suppress and eventually to eradicate *B. hilaris* on the island of Pantelleria. Understanding the mechanism of sperm competition in this species may help to manage and organize the eventual application of the Sterile Insect Technique. Indeed, it is essential that irradiated males are able to sexually compete against non-irradiated males, both in mating with females and in fertilizing their eggs.

## 2. Materials and Methods

### 2.1. Insect Collection and Rearing

Individuals at different phenological stages (mainly adults, but also late (4th and 5th) instar nymphs) of *B. hilaris* were collected from infested caper plants at the “Cooperativa Agricola Produttori Capperi”, Contrada Scauri Basso, Pantelleria, TP (GPS coordinates 36°46′23″ N 11°57′41″ E), during the summers and autumns of 2019 and 2020. Collections were carried out using a small entomological vacuum device connected to a 100 cc transparent plastic bottle. A careful inspection of each individual caper plant (stems and leaves), the plot stone walls, around, and the ground below the plant, allowed for a mass trapping of bagrada bug individuals, especially during the late summers and throughout autumns, when the aggregation behavior is obvious [58]. Individuals were transported in cardboard tubes (4.1 × 11.9 cm).

At the laboratory, the collected bagrada bugs were placed into several cubic sleeve cloth cages (BugDorm^®^, Taichung, Taiwan) (60 × 60 × 60 cm, 680 µm mesh opening), in order to set up a small laboratory colony at the quarantine facility of the Edmund Mach Foundation (FEM), San Michele (TN), Italy. Inside each cubic cage were placed paper sheets at the bottom, to maintain the correct moisture content, and one open plastic Petri dish (9 cm diameter) with fine sand (granulometry = 200 µm) to provide oviposition sites. The controlled climatic conditions were set with temperatures ranging from ca. 22 °C (night) to ca. 26 °C (day); light was supplemented with sodium lamps (L/D 14:10; RH: 50–60%). *Brassica oleracea* L. var. *gemmifera* (Brussels sprout) was placed as food source.

Each cage was cleaned and the food replaced three times a week to always maintain a fresh food source.

### 2.2. Irradiation

Fifth-instar nymphs were taken from the rearing cages using plastic 50 mL CorningTM Falcon tubes. Subsequently, single individuals were isolated in 5 cm diameter Petri dishes (with a small leaf of Brussels sprout as food source) until they reached adulthood. Then, males were kept separated from females in two different cages, according to their sexual dimorphism [21]. This procedure allowed for ensuring the virginity of newly emerged individuals. Groups of 10 males were then placed in 9 cm diameter Petri dishes and irradiated at three different doses (60, 80, and 100 Gy) at the Calliope Facility of ENEA Casaccia Research Centre (Rome) [59]. The dose rate was 175.03 Gy/h (2.92 Gy/min). The irradiation of males at 60 Gy was performed on 10/28/2022, at 80 Gy on 10/13/2022, and at 100 Gy on 18/11/2022. Non-irradiated females and males, to be used as positive controls, were also placed under the same controlled conditions as the irradiated individuals.

### 2.3. Experimental Set-Up

In order to detect the possible presence of sperm selection mechanisms, a double mating scheme was used, confining in no-choice conditions a virgin female individually with a non-irradiated or irradiated male (Co-60), followed by with an irradiated or non-irradiated male, respectively [46]. Males were irradiated at three different doses: 60, 80, and 100 Gy. Two bioassays were then set up, by placing one pair of individuals in a 500 mL transparent glass jar covered with a 680 µm white polyester mesh, with a 5 cm diameter open Petri dish filled with fine sand as an oviposition site.

In treatment A, the first male with whom the female was confined was non-irradiated. At the end of day 15, the non-irradiated male was removed and replaced with an irradiated male (at 60, 80, or 100 Gy) for 15 days (non-irradiated–irradiated order). In treatment B, the same experimental design was carried out, but following a reverse order (irradiated–non-irradiated order). For both treatments, ten replicates were performed for each of the three irradiation doses (Figure A1, Appendix A).

Each pair was fed with a single Brussels sprout. The experiments started on the same day as the irradiation (see above) and ended after 30 days.

For the controls, the tests were set up as follows:-a non-irradiated–non-irradiated mating order, i.e., two non-irradiated males to verify the reproductive parameters (10 replicates) as positive control (Figure A2, Appendix A);-an irradiated–irradiated mating order (10 replicates) i.e., two irradiated males as negative control (Figure A2, Appendix A);-a female and a non-irradiated male, without male replacement (10 replicates) (Figure A2, Appendix A);-a single female virgin, to consider the number of eggs laid regardless of mating (10 replicates) (Figure A2, Appendix A).

The following experimental procedures were performed in order to compute and record fecundity (number of eggs laid), fertility (number of emerged nymphs), and the percentage of hatched eggs for each treatment and control. Every five days, the sand in the Petri dishes was inspected under a stereomicroscope (Olympus SZX-ILLB200; Hamburg, Germany) at 20× magnification, recording the numbers of laid eggs. The collected eggs were transferred, using a fine paint brush, to a filter paper housed in a 3 cm diameter Petri dish and kept under the same climatic conditions as described above, in order to estimate the eclosion rate. Eggs were considered non-viable if they did not hatch any progeny after two weeks.

At the end of day 15 from the start of the experiment, the first males were removed. This temporal choice was decided upon based on previous biological observations: we noted that, during the first five days, very few eggs hatched from a female mating with a non-irradiated male. On the other hand, a further extension, i.e., adding more than 15 days to prolong the experimental phases, could be more problematic for the onset of mortality factors among the individuals.

All experiments were carried out under the same laboratory conditions as the colony rearing (described above).

### 2.4. Data Analysis

A glmer analysis (generalized mixed model with random effects, in the statistical environment R) [60,61] was performed on three reproductive parameters (the variables of response): the number of emerged nymphs per observation, the number of eggs oviposited per observation, and the percentage of hatched eggs from the total of laid eggs per observation. A fixed factor was considered for each dose, i.e., the type of treatment, corresponding to four levels (first mating with non-irradiated male, second mating with irradiated male—as in treatment A—first mating with irradiated male, and second mating with non-irradiated male—as in treatment B) (see the experimental scheme reported in Appendix A). The level “mating before with non-irradiated male” was set to zero in the model and treated as a reference variable. A glmer model was also applied to the non-irradiated control in order to compare the irradiation effects with the effects that are due to a natural decay in reproductive parameters over time. Thus, in the control-only model, reproductive parameters were compared between the second and the first mating with the non-irradiated male. The female used in the experiment was considered as the random effect. The Tukey’s test for mean separation was performed using the multcomp package [62] in the statistical environment R, on the four types of treatment calculated within each dose applied. Due to overdispersion in the data, negative binomial distributions were applied for the number of emerged offspring and for the number of laid eggs, whereas a gamma distribution was applied for the percentage of hatched eggs [63]. In the latter case, a coefficient of 1 was added to the outcome variable because the parameters of the gamma distribution must always be positive. A linear mixed model (lmer package in the R environment) was applied to the P2 values, as a normal distribution was assumed for the outcome variable P2. The P2 index can be used to estimate the proportion of sperm transmitted from the second male, using irradiated males in the studies of last male sperm precedence [46]. P2 values indicate a proportion (from 0 to 1) and are calculated through the estimation of the changes in the percentage of eggs hatching, due to the effect of the second mating in the order irradiated male–non-irradiated male or non-irradiated male–irradiated male. In the case of sperm transmission from the irradiated male (detected as a decrease in fertility values), the proportion of the transmitted sperm can be expressed in the reciprocal form 1-P2. P2 values were computed as described by Sillen-Tullberg [64]. In this case, the female was the random effect, while the applied dose of irradiation and the type of male replacement (non-irradiated or irradiated) were considered as fixed effects. The type of replacement was nested into the dose treatment in order to point out differences in the proportion of sperm transmission between the non-irradiated male replacement (the reference variable) and the irradiated male replacement.

## 3. Results

### 3.1. Effect of the Last Male on the Number of Oviposited Eggs

The means and standard errors of all experiments are reported in Table A1 of the Appendix A. In general, physiological differences were observed among the three experiments at the three tested doses. In particular, the females who mated for the first time with the non-irradiated male laid more eggs in the experiments at 60 Gy (29.58 ± 2.16) and 80 Gy (38.30 ± 2.93) than in the experiment at 100 Gy (20.63 ± 3.03). The irradiation of the male negatively affected the number of oviposited eggs. This was also evident in the first mating, when any physiological decay in the oviposition trend had yet occurred (23.41 ± 1.58 eggs on the average, against 30.39 ± 1.71 eggs recorded on the females mated with untreated males). At 60 Gy (Table 1, Figure 1, a statistically significant decrease in the rate of oviposition was observed when the non-irradiated male was replaced with the irradiated one (coef = −0.451, z-value = −2.690, *p*-value = 0.0072; mean from 29.58 ± 2.16 to 19.14 ± 2.13). The drop in the number of oviposited eggs was highly significant and even stronger at 80 Gy (from 38.30 ± 2.93 eggs to 13.73 ± 1.55). Regarding the opposite order of mating, at 60 Gy, the number of laid eggs slightly decreased after the replacement of the irradiated male with the non-irradiated one (from 27.62 ± 2.47 to 23.62 ± 1.80 eggs). However, this decrease was not statistically distinguishable from the values obtained in the first mating with the non-irradiated male (coef = −0.242, z-value = −1.299, *p*-value = 0.219). Instead, a significant drop was observed at 80 Gy (from 27.43 ± 2.65 to 15.33 ± 1.94 eggs; coef = −0.9155, z-value = −5.029, *p*-value ≤ 0.0001). It is worth noting that the changes in the oviposition rate were not obvious at 100 Gy for all treatments (from 23.63 ± 3.03 to 16.18 ± 1.88 in the replacement with the irradiated male and from 13.93 ± 2.42 to 14.37 ± 1.81 in the replacement with the non-irradiated male), and no statistically significant differences were detected (Table 1, Figure 1).

### 3.2. Effect of the Last Male on the Number of Newly Hatched Nymphs

As Table 2 of the fixed effects shows, a strong negative effect of the male irradiation on the number of newly hatched nymphs was observed when the first male was irradiated at all the tested doses (Table A1), and particularly at the dose of 100 Gy. A reduction in eggs hatching, in comparison to the hatching rate obtained in the first mating with the untreated male, was also detected when the female was mated with a second male, regardless of whether the male was irradiated or not. In the untreated control with the non-irradiated–non-irradiated male replacement (Table A1 in the Appendix A), the number of eggs decreased, from 19.48 ± 3.76 from the first mating to 13.76 ± 1.85 from the second mating, and the number of offspring decreased from 10.71 ± 2.87 to 6.57 ± 1.47; thus, a weaker reduction was observed, in comparison to the drop from 29.1 ± 3.52 to 16.35 ± 2.64 eggs and from 17.2 ± 1.36 to 5.71 ± 0.66 hatched eggs observed in the second mating with irradiated males. In detail, (Figure 2), an apparent suppression in the number of offspring was observed at the 60 Gy dose when the non-irradiated male was replaced with the irradiated one (coef = −0.767, z-value = −2.265, *p*-value = 0.0235, mean from 15.27 ± 1.99 to 6.32 ± 1.30). Conversely, only a partial recovery of fertility was observed at 60 Gy when the irradiated male was replaced with the untreated male (Figure 2), but in this case, the mean values did not significantly differ from those obtained in the first mating with the non-irradiated male (mean from 1.54 ± 0.50 to 7.59 ± 1.04 emerged nymphs, against the 15.27 ± 1.99 emerged nymphs recorded during the first mating). The difference among the treatments was even stronger in the case of irradiation at 80 Gy, with a clear suppression in the number of emerged nymphs when the untreated male was replaced with the irradiated one (mean from 22.82 ± 2.55 to 6.30 ± 1.12). In the reverse order, only a weak recovery was observed when the irradiated male was replaced with the non-irradiated male (mean from 3.53 ± 1.14 to 8.57 ± 1.58). At 80 Gy, the fertility values obtained in the second mating were both significantly lower than those recorded in the first mating with the non-irradiated male, regardless of whether the second male was irradiated or not (Table 2, Figure 2). At the 100 Gy dose, the decrease in fertility after the replacement with the irradiated male was strong, and only a few eggs hatched (mean from 12.21 ± 1.96 to 4.39 ± 1.00). In addition, the recovery of fertility was poor after replacement with the non-irradiated male. These values were considerably lower than those obtained in the first mating with the non-irradiated male (from 0.30 ± 0.17 to 2.87 ± 1.09 emerged nymphs, against the 12.21 ± 1.96 emerged nymphs recorded with the untreated male; see Table 2 for comparisons, and Table A1).

### 3.3. Effect of the Last Male on the Percentage of Hatched Eggs

As expected, the strongest negative effect of mating with an irradiated male on the percentage of hatched eggs was found in the first mating (Table 3 and Table A1 in the Appendix A), whereas a significant drop in eggs hatching after replacement with an irradiated male occurred only in the case of mating with the irradiated male at 100 Gy (coef = −1.090, z-value = −2.764, *p*-value = 0.0057; means from 57.38 ± 4.67 to 21.60 ± 3.86). In fact, the effect of suppression on egg hatching was significantly weaker at 60 Gy than at 100 Gy, and even weaker at 80 Gy (Figure 3). At the 60 Gy dose, the percentage of hatching decreased from 49.12 ± 5.60% to 30.87 ± 5.41%, which was not significantly different from the values obtained with the first untreated male. At the 80 Gy dose, the second mating with the irradiated male contributed even less to lowering the hatching rate (from 59.60 ± 5.27% to 43.99 ± 5.94%). Conversely, a significant recovery in the hatching rate was observed at 60 Gy when the irradiated male was replaced with the non-irradiated one (from 5.31 ± 1.56% to 31.74 ± 3.96%, against the 49.12 ± 5.60% recorded in the first mating with the untreated male). An almost-complete recovery in the percentage of hatched eggs occurred at 80 Gy when the irradiated male was replaced with the non-irradiated one (from 10.42 ± 2.73% to 56.05 ± 6.53% against a hatching rate of 59.60 ± 5.27% recorded with the first untreated male; see Figure 4 for trends). Thus, the percentage of eggs hatching was almost completely restored at 80 Gy, after the second mating with the non-irradiated male, indicating that the decrease in reproductive parameters was principally due to a drop in the rate of oviposition. On the other hand, in the experiment at 100 Gy, only a slight recovery in the percentage of fertile eggs was observed after the second mating with the non-irradiated male (from 3.39 ± 1.84% to 13.93 ± 4.31%; see Figure 3 and trends in Figure 4). The latter value was much lower than that recorded for females mated for the first time with non-irradiated males (13.93 ± 4.31% against 57.38 ± 4.67%, respectively). Thus, a strongly significant suppression of this reproductive parameter was evident in both mating orders at 100 Gy (Table 3, Figure 3, Table A1 and Figure 4 for trends).

### 3.4. P2 estimates of the Sperm Transmission from the Last Male

The P2 values represent the results for the reproductive parameters analyzed here. As shown in Table 4, the overall P2 values were significantly lower at the dose of 100 Gy, in comparison with the 60 Gy dose (coef. = −0.43; *t* value = −2.65; *p*-value = 0.0137), while a trend towards higher values was observed only for the treatment at 80 Gy. Regarding the order of replacement, a moderate imbalance was observed at the 60 Gy dose between the second irradiated male and the second non-irradiated male (0.646 ± 0.08 in the replacement from irradiated to non-irradiated, and 0.371 ± 0.11, in the opposite order), and no appreciable differences were observed from a statistical point of view. An even stronger and more significant imbalance towards the non-irradiated male was observed at 80 Gy (coef. =−0.69; *t* value = −4.06; *p*-value = 0.00015), with a mean of 0.935 ± 0.12 in the replacement from an irradiated male to a non-irradiated male, and a mean of 0.286 ± 0.11 in the replacement from the non-irradiated male to the irradiated one. In contrast, a clear imbalance for the irradiated male was evident at a dose of 100 Gy (coef. = 0.41; *t* value = 2.38; *p*-value = 0.0209), with a mean of 0.222 ± 0.08 when the irradiated male was replaced with the non-irradiated male, and a mean of 0.641 ± 0.07 in the replacement from the non-irradiated male to the irradiated male. The P2 values recorded at different doses, and with different orders of mating, are shown in Figure 5.

## 4. Discussion

This study is the first to investigate the mechanism of sperm competition in *B. hilaris*. We observed that the experiment in which females mated with irradiated males first reduced the number of offspring produced. It is worth noting that part of the reduction in the number of emerged nymphs (Table A1 in the Appendix A) could be attributed to a physiological decrease in fecundity over time, but the decrease was stronger than that due to the natural trend. In the non-irradiated–irradiated mating order, no precedence for the last male was recorded, as the hatching percentage of the eggs was lowered, but not as when a virgin female was only exposed to an irradiated male. With regard to the irradiated–non-irradiated mating order, it can be seen that, as soon as the female was placed with a non-irradiated male, the percentage of hatching eggs increased, but did not return to control values. There was no complete adoption of the last male sperm precedence mechanism because, after replacing irradiated males with non-irradiated ones, fertility was not fully recovered (irradiated–non-irradiated order); moreover, in the reverse order (non-irradiated–irradiated), an immediate decline in fertility was not recorded. This suggests that sperm from both males appear to mix in the spermatheca, indicating that *B. hilaris* has a sperm-mixing mechanism. This was also confirmed by the P2 values. In insects, there are two predominant patterns of sperm utilization: P2 values between 0.4 and 0.7 indicate species in which the sperm of the two males are mixed, resulting in sperm competition; whereas P2 values above 0.8 indicate mechanisms of sperm precedence for the second male or sperm displacement [51]. At 60 Gy, the values obtained indicated a potential mixing of the sperm; in fact, in the non-irradiated–irradiated order, the P2 value was approximately 0.40, whereas in the irradiated–non-irradiated order, it was approximately 0.60. At an intermediate dose, i.e., 80 Gy, there is a preference for the non-irradiated male, regardless of the mating order; in fact, when a female is placed with an irradiated male first, the P2 value is approximately 1, whereas, when the treated male is placed second, the P2 value drops to 0.28. Our results show that a female prefers the sperm of the non-irradiated male at the 60 and 80 Gy doses. At these irradiation doses, the irradiated sperm is less competitive. On the other hand, we observed the opposite mechanism at 100 Gy, at which the female showed a clear preference for the sperm of the irradiated male, as confirmed by the P2 values (see Figure 5), which, in the irradiated–non-irradiated order, had a particularly low value (0.22). This result indicates the presence of sperm utilization in favor of the sperm of 100 Gy-irradiated males. The high variability in the P2 values can be explained by sperm viability. Indeed, this appears to be higher in polyandrous species than in monandrous species [65]. In addition, high intraspecific variation in the proportion of dead sperm has been observed [65]. The low P2 value obtained at 100 Gy can be interpreted as a consequence of irradiation. Irradiated *B. hilaris* males may have suffered a reduction in sperm number, motility, and/or viability as a result of gamma-ray exposure. Indeed, in *N. viridula* males irradiated at 40 Gy, abnormalities are present in axonemes, mitochondrial derivatives, the nebenkern, and centrioles [66]. In order for sperm competition to occur, live sperm from irradiated and non-irradiated males must coexist within the female reproductive tract [65]: in *B. hilaris*, the consequences for a female mating first with an irradiated male at 100 Gy could prevent the entry of non-irradiated male sperm into the spermatheca, or create hostile physiological environmental conditions unsuitable for their survival. In insects, the irradiation process may negatively impact the length of the sperm and cause the blockage of storage organs [67,68] as they move slowly and differently [69,70]. Within Heteroptera, sperm dimorphism is frequently present [71], and the irradiation process may have promoted the presence of long sperm, which prevented the entry of non-irradiated sperm into the spermatheca. This may have induced the sperm utilization for 100 Gy irradiated males. In Sterile Insect Technique programs, irradiated males often show a lower capacity to fertilize than non-irradiated males [51]. However, this pattern is not always valid. Irradiated males exhibited a better fertilizing capacity than normal males in tests of the armyworm *Pseudaletia separata* (Walker) and the melon fly *Bactrocera dorsalis* Hendel [72,73].

Irradiation may also induce changes in courtship behavior. Pentatomids use vibrations, transmitted by substrates, for short-range courtship [74]. It has been observed that *B. hilaris* males generate pulse trains with a harmonic frequency structure during courtship. A similar signal is used during copulation [75]. In a recent study [48], the differences between irradiated (at 60 and 100 Gy) and non-irradiated *B. hilaris* males were analyzed at the level of vibrational communication before and during copulation. The results showed that males irradiated at 60 Gy had similar vibrational signal frequencies to the control, and comparable mating successes. However, at 100 Gy, they emitted signals with lower peak frequencies and mated less than the control. Thus, following irradiation at low doses (60 Gy), males were highly competitive with their wild counterparts. In addition, fecundity did not differ significantly between the two mating orders. In fact, the number of eggs laid was not altered by the change in the males. This result is very promising, in view of the integration of classical biological control with the Sterile Insect Technique. In fact, if an oophagous parasitoid of *B. hilaris* is released, it could also use eggs fertilized by an irradiated male to promote parasitoid reproduction. Choice tests are necessary in order to verify that the chosen parasitoid utilizes both non-irradiated eggs and those produced using the sperm of the irradiated male equally, which is under evaluation (RFH Sforza, personal communication). In the negative control, when a female was confined with two irradiated males, there was a further decline in fertility (Appendix A). This latter result is highly promising from the perspective of a SIT program with massive field releases of sterile adults, as the probability of a female mating with more irradiated males is higher than that of mating with wild non-irradiated males.

The mechanism of sperm mixing recorded in *B. hilaris* can be a function of numerous morphological and behavioral traits. Among these, mate guarding is the most widespread phenomenon in insects [76]. Prolonged mate guarding prevents the precedence of sperm [77,78,79]. In *B. hilaris*, in order to achieve a chance to mate with a female, competing males are frequently spotted around copulating pairs, and may even be located on top of them (personal observations). Therefore, a prolonged copulation time may be a mate-guarding strategy [21]. The physical characteristics of the female sperm storage organ may affect the degree of non-random paternity [80]. The spermathecal form determines how much sperm would mingle within the female sperm storage organ [51]. The last male sperm to enter the female storage organ would be the first to leave, resulting in last-male sperm precedence in species with complicated, elongated, or tubular spermathecae. The probability of sperm mixing and mixed paternity should be higher in species with simple spherical spermathecae [51]. The morphology of the spermatheca of *B. hilaris* appears to be similar to the characteristics of the second group [47]. In addition, the utilization of sperm is also influenced by variations in the mixing potential of ejaculates, which is determined by physical characteristics, such as the viscosity of the ejaculate, motility of the sperm, or the space available within the sperm storage organs [51]. Understanding the mechanisms of sperm transfer and storage methods is necessary in order to confirm sperm competition [81]. Among the Pentatomidae family, the data reported on *N. viridula* provided evidence in favor of sperm-mixing, sperm stratification, and sperm displacement hypotheses [82]. Therefore, the initial sperm-mixing hypothesis for *B. hilaris* does not seem to be far from the sexual competition mechanisms present in its family. Additional research on the *B. hilaris* mating system is required to further understand the mechanism of sperm mixing. This research should consider a variety of factors, such as the timing, number, duration, and frequency of copulations, as well as the lifetime and fitness of the sperm [50,51].

The application of the Sterile Insect Technique on *B. hilaris* can be considered a viable strategy. It is essential to consider how *B. hilaris* biology may affect its implementation, and how this may differ from SIT programs in Diptera and Lepidoptera, for which the technique is well established [83,84]. Adults must be released in masses for SIT application: in Lepidoptera and Diptera, the mass release of sterile adults is not a problem because they cause the most damage at the larval life stage [85]. The situation is different in Hemiptera, for whom the release of adult stages generally leads to an increase in damage [35,37]. However, in *B. hilaris*, males were found to feed less than females, and during long mating acts, male feeding is neglectable [86]; consequently, mass release would not produce a significant amount of damage. Polyandry may threaten the success of the SIT [87] because wild females can mate with both non-irradiated and irradiated males [88]. This problem can be addressed through a release that allows for an overabundance ratio of sterile males [38,88]. Since there is concern regarding the suitability of setting up expensive large mass-rearing facilities for Hemiptera pest species [89,90,91], we are considering the possibility of developing small-scale SIT programs on the bagrada bug, combining mass trapping and SIT strategies: using the peculiar behavior of this pentatomid bug species to collect large numbers of specimens in the field during the autumn aggregation phase, sex them, preserve the females (for laboratory rearings), and then proceed to irradiate wild males for their consequent release in the field. Irradiated males released at the same sites where the mass trapping was carried out would find fewer fertile males with whom compete, and, consequently, they will have more chances to mate with the remaining wild females. Large-scale sexing methods have not yet been developed, but size/shape differences between males and females could be used for this purpose. Mass-release methods have not yet been developed.

The most suitable irradiation dose will be selected based on the results of further experiments. Based on recent studies, the most suitable doses are 60 and 80 Gy. In fact, although there is a “blockage” in the spermathecae that prevents the reception of additional sperm at 100 Gy, additional factors should be taken under consideration. At this dose, males were not found to be competitive in the study of vibrational communication [48], in contrast to those subjected to the 60 Gy dose. The model proposed by Parker and Mehta [34] for determining the optimal dose for SIT application involves studying the correlations between dose and both fertility and competitiveness. The relationship with fertility is already known [30], so future studies should focus on the competitiveness of males irradiated at different dose rates. The correct application of the SIT is based on the precise ratio of irradiated males that should be released in order to overcome the wild-type fertile males. The number of males released is based on the size of the wild population and the performance of irradiated males released in the field [49]. A minimum irradiated male:non-irradiated male ratio of 10:1 has been shown to have an effect on wild population reduction [92]. If the production of sterile males allows for higher overfeeding ratios, the application of the SIT will be even more effective [44]. It will then be necessary to conduct experiments first in the laboratory, where different ratios of irradiated:non-irradiated individuals will be tested in order to identify which number of individuals is most suitable for release. Field trials will then be conducted. In addition, dispersal studies are required, and it is critical that sterile males are distributed throughout the release area [93,94]. The frequency of sterile insect releases differs among species and varies with their average longevity [35]. In *B. hilaris*, no differences in longevity were observed between non-irradiated and irradiated individuals [30]. Irradiated individuals should be released during seasonal periods when the population is declining [92]. In addition, mating is concentrated at specific times of the year, and the production and release of sterile insects will need to be timely in order to ensure maximum benefits [95].

It is reasonable to evaluate the suitability of a SIT strategy as a potential component of an area-wide integrated pest management program, particularly in well-isolated and restricted locations, in light of the recent literature records and our findings.

## 5. Conclusions

This study on sperm competition in *B. hilaris* revealed the presence of sperm mixing. At 60 and 80 Gy, the non-irradiated male is favored, in contrast to the highest dose (100 Gy), at which the irradiated male dominates. As a result, the sperm performance of irradiated males in no-choice and choice conditions needs to be further investigated.

Furthermore, it was found that irradiated males can partially reduce the fertility of females, regardless of the mating order. Additional behavioral bioassays, also in more gregarious contexts, for evaluating the mating performance of irradiated adults of *B. hilaris* in both no-choice and choice conditions will be of fundamental importance for the application of the Sterile Insect Technique in open field conditions.

Finally, pre-release tests will be carried out under semi-field conditions with small-scale populations, in order to estimate the optimal number of irradiated individuals needed to achieve a decline in fertility.

Behavioral and histological investigations will also be essential, in order to highlight any differences between irradiated and non-irradiated males and their eventual mating effects on the spermatheca structure.

## Figures and Tables

**Figure 1 insects-14-00681-f001:**
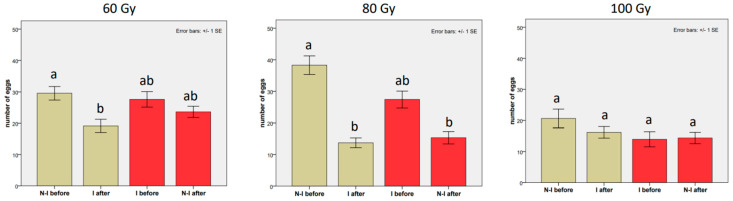
From left to right: Effects of the last male of *Bagrada hilaris* on the number of laid eggs at 60, 80, and 100 Gy. Grey histograms: a non-irradiated male was replaced with an irradiated male (treatment A). Red histograms: an irradiated male was replaced with a non-irradiated male (treatment B). “NI−I” refers to the non-irradiated male; “I” refers to the irradiated male. Bars with no common letters are significantly different at *p* < 0.05.

**Figure 2 insects-14-00681-f002:**
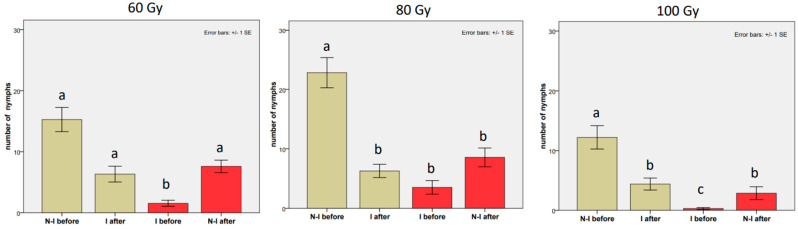
From left to right: Effects of the last male of *Bagrada hilaris* on the number of offspring at 60, 80, and 100 Gy. Grey histograms: a non-irradiated male was replaced with an irradiated male (treatment A). Red histograms: an irradiated male was replaced with a non-irradiated male (treatment B). “N−I” refers to the non-irradiated male; “I” refers to the irradiated male. Bars with no common letters are significantly different at *p* < 0.05.

**Figure 3 insects-14-00681-f003:**
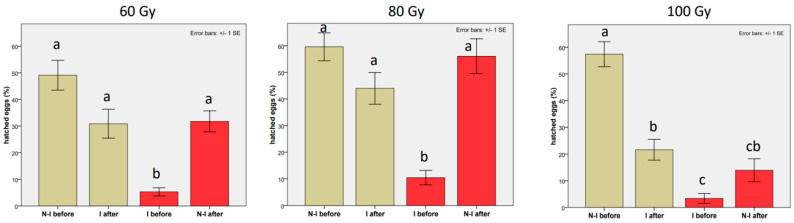
From left to right: Effects of the last male of *Bagrada hilaris* on the percentage of hatched eggs at 60, 80, and 100 Gy. Grey histograms: a non−irradiated male was replaced with an irradiated male (treatment A). Red histograms: an irradiated male was replaced with a non−irradiated male (treatment B). “N−I” refers to the non-irradiated male; “I” refers to the irradiated male. Bars with no common letters are significantly different at *p* < 0.05.

**Figure 4 insects-14-00681-f004:**
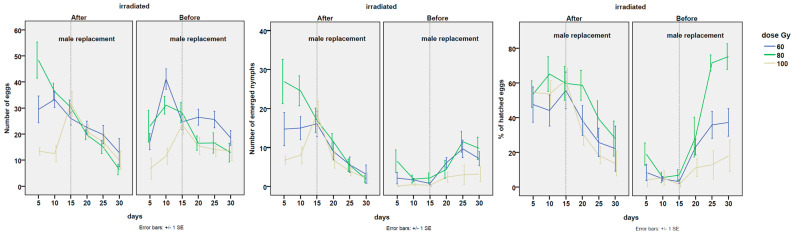
From left to right. Trends in the number of eggs, number of emerged nymphs, and percentage of eggs hatching of *Bagrada hilaris* during the experiment on sperm selection at the 3 tested doses of 60, 80, and 100 Gy. After 15 days, the male was replaced. In the panel on the left, the non−irradiated male was replaced with the irradiated one (treatment after). In the panel on the right, the irradiated male was replaced with the non−irradiated one (treatment before).

**Figure 5 insects-14-00681-f005:**
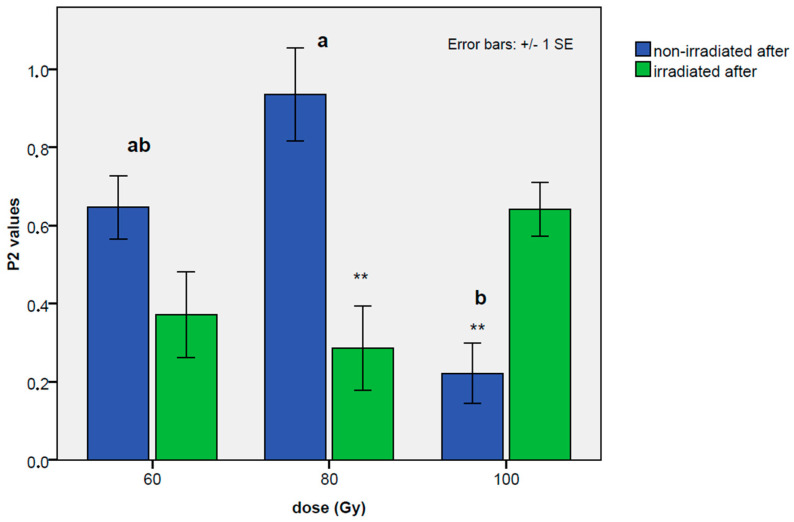
P2 values for the second mating with the non-irradiated male (blue histograms, treatment B) or with the irradiated males (green histograms, treatment A) at the tested doses of 60, 80, and 100 Gy. Bars with no common letters indicate a statistically significant difference at *p* = 0.05, according to the Tukey’s test for mean separation among the different doses for the type of treatment (non-irradiated after or irradiated after). The asterisks indicate a statistically significant difference between the second mating with a non-irradiated male and the second mating with an irradiated male within each dose treatment. No letters or asterisks indicate no statistically significant differences.

**Table 1 insects-14-00681-t001:** Glmer model applied to the dependent variable “number of eggs” of *Bagrada hilaris* at the three tested irradiation doses 60, 80, and 100 Gy.

Experiment	Fixed Effects	Estimate	S.E.	z Value	*p*-Value
	(Intercept)	3.33871	0.14388	23.204	<2 × 10^−16^ ***
	Irradiated before	−0.04768	0.19401	−0.246	0.80588
60 Gy	Non-irradiated after	−0.24183	0.19680	−1.229	0.21913
	Irradiated after	−0.45079	0.16758	−2.690	0.00715 **
	(Intercept)	3.6455	0.1231	29.606	<2 × 10^−16^ ***
	Irradiated before	−0.3338	0.1794	−1.860	0.0628
80 Gy	Non-irradiated after	−0.9155	0.1821	−5.029	4.94 × 10^−7^ ***
	Irradiated after	−1.0261	0.1782	−5.760	8.43 × 10^−9^ ***
	(Intercept)	3.0265	0.1966	15.395	<2 × 10^−16^ ***
	Irradiated before	−0.3922	0.2652	−1.479	0.139
100 Gy	Non-irradiated after	−0.3616	0.2651	−1.364	0.173
	Irradiated after	−0.2428	0.2688	−0.903	0.366
	(Intercept)	2.8598	0.2770	10.32	<2 × 10^−16^ ***
Untreated control	2nd non-irradiated	−0.3341	0.3930	−0.850	0.395

Significance codes: ≤0.001, ***; ≤0.01, **.

**Table 2 insects-14-00681-t002:** Glmer model applied to the dependent variable “number of emerged nymphs” for *Bagrada hilaris* at the three tested irradiation doses 60, 80, and 100 Gy.

Experiment	Fixed Effects	Estimate	S.E.	z Value	*p*-Value
	(Intercept)	2.2384	0.3443	6.501	8.00 × 10^−11^ ***
	Irradiated before	−2.2593	0.4855	−4.653	3.26 × 10^−6^ ***
60 Gy	Non-irradiated after	−0.5793	0.4658	−1.244	0.2136
	Irradiated after	−0.7673	0.3387	−2.265	0.0235 *
	(Intercept)	3.1276	0.2226	14.049	< 2 × 10^−16^ ***
	Irradiated before	−1.8653	0.3347	−5.573	2.51 × 10^−8^ ***
80 Gy	Non-irradiated after	−0.9797	0.3263	−3.002	0.00268 **
	Irradiated after	−1.2865	0.3203	−4.017	5.90 × 10^−5^ ***
	(Intercept)	2.3465	0.3777	6.212	5.22 × 10^−10^ ***
	Irradiated before	−4.1015	0.6758	−6.069	1.29 × 10^−9^ ***
100 Gy	Non-irradiated after	−2.1291	0.5757	−3.698	0.000217 ***
	Irradiated after	−1.1008	0.5290	−2.081	0.037455 *
	(Intercept)	1.8527	0.4317	4.291	1.78 × 10^−5^ ***
Untreated control	2nd non-irradiated	−0.3559	0.4428	−0.804	0.422

Significance codes: ≤0.001, ***; ≤0.01, **; ≤0.05, *.

**Table 3 insects-14-00681-t003:** Glmer model applied to the dependent variable “percentage of hatched eggs” of *Bagrada hilaris* at the three tested doses 60, 80, and 100 Gy.

Experiment	Fixed Effects	Estimate	S.E.	z Value	*p*-Value
	(Intercept)	3.8963	0.2227	17.495	<2 × 10^−16^ ***
	Irradiated before	−2.2073	0.3061	−7.211	5.54 × 10^−13^ ***
60 Gy	Non-irradiated after	−0.4355	0.3103	−1.403	0.161
	Irradiated after	−0.4633	0.3402	−1.362	0.173
	(Intercept)	4.0894	0.2339	17.485	<2 × 10^−16^ ***
	Irradiated before	−1.7362	0.3362	−5.164	2.42 × 10^−7^ ***
80 Gy	Non-irradiated after	−0.0613	0.3493	−0.175	0.861
	Irradiated after	−0.30314	0.3424	−0.885	0.376
	(Intercept)	4.0537	0.3755	10.794	<2 × 10^−16^ ***
	Irradiated before	−3.4021	0.5906	−5.760	8.39 × 10^−9^ ***
100 Gy	Non-irradiated after	−1.9794	0.7026	−2.817	0.00484 **
	Irradiated after	−1.0899	0.3943	−2.764	0.00571 **
	(Intercept)	0.02352	0.00745	3.158	0.00159 **
Untreated control	2nd non-irradiated	0.00094	0.00603	0.155	0.87648

Significance codes: ≤0.001, ***; ≤0.01, **.

**Table 4 insects-14-00681-t004:** Linear mixed model applied to the P2 values found for the second male. The fixed effects, dose and type of second mating, are reported. The effect of the second male (irradiated or non-irradiated) is computed within each dose treatment. The re-mating with the non-irradiated male is the reference level.

	Estimate	S.E.	*t* Value	*p*-Value
(Intercept)	0.6492	0.1090	5.956	2.04 × 10^−7^ ***
80 Gy	0.2879	0.1637	1.759	0.08408
100 Gy	−0.4323	0.1629	−2.654	0.01037 *
Dose 60: irradiated male	−0.2477	0.1636	−1.514	0.13513
Dose 80: irradiated male	−0.6857	0.1691	−4.055	0.00015 ***
Dose 100: irradiated male	0.4067	0.1711	2.377	0.02090 *

Significance codes: ≤0.001, ***; ≤ 0.05, *.

## Data Availability

The data presented in this study are available on request from the corresponding author.

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
