# Peer review of "Using Gamma Irradiation to Predict Sperm Competition Mechanism in Bagrada hilaris (Burmeister) (Hemiptera: Pentatomidae): Insights for a Future Management Strategy"

_insects, 2023, doi:10.3390/insects14080681_

Round 1

Reviewer 1 Report

Simple summary:

L17- The opening sentence is not convincing. It should be like “the greatest challenge for modern agriculture is the insect pest damage, and insecticides are the biggest tool used for controlling these insects”.

L23: Should not use colon.

Abstract:

L26: Brassica crops, no need to mention species.

L27: Keep consistency of tense (either present or past tense, not mixed them up).

L28: For agricultural pest control, better not use the word like eradication, which is not practically possible. Instead, use word like mitigation or minimise the pest severity or alleviation etc.

Overall: Abstract is not very comprehensive. In my understanding, three doses of cobalt irradiation are used considering three different parameters such as fecundity, fertility, and egg hatching. But the clarity is lacking to pinpoint the experiments. There is an ample root for the improvement in terms of clear thinking and writing.

Introduction:

In sum up, the introduction is very exhaustive which does not necessarily need too much information. Once you have defined the wild population as the non-irradiated males, be consistent with terminology either wild male or non-irradiated males or control.

The objectives should be also clearly defined. Also, briefly project the result implication or application for the future endeavour.

Materials and Methods:

This section is meant to be succinct and clear. And the flow of activities should be maintained to make it reader friendly. The subsection (L188) is too verbose. No necessary to further explanation of worthiness of methods they use in this section, which could be worthwhile while incorporating in discussion section.

Results:

This section also needs clear and concise writing. Needed to maintain coherence in writing.

Conclusion:

The authors should pitch out the research findings here. And further pointing out the research implications based on this finding. In this section, the authors started with recommendations without emphasizing their own work. Therefore, this section also demands substantial rewriting to make this section readable.

Overall feedback:

This is a good piece of work, on where the authors were able to establish good set of data on insect control using sterile insect technology. However, in order to make it publishable, significant improvement is essential, especially in rewriting. There is not any alignment between the study objectives and materials and methods. In conclusion, the manuscript is not deemed to be publishable as it is or with minor correction. Following points are prescribed to make it worthy for the publication and these include substantial improvement in English rewriting, making the manuscript succinct, and maintaining the writing flow.

Extensive linguistics rewriting is essential.  I found grammatical errors and poor structure in many places which are avoidable. 

Reviewer 2 Report

Review of: Using Gamma Irradiation to Predict Sperm Competition Mechanism in Bagrada hilaris (Burmeister) (Hemiptera: Pentatomidae): Insights for a Future Management Strategy

By Chiara Elvira Mainardi et al.

This paper is an interesting contrast to many similar papers that I review and read in that polyandry is typical. From the standpoint of SIT, this is not a problem as, in contrast to Knipling's assumptions, remating is not prohibitive and can even be beneficial - citations to this effect are included.

The experimental designs are reasonable and appear to have been carried out logically and consistently. A very good job.

I have made suggestions for wording and clarity in the marked-up PDF.

The authors might want to spend a bit of time briefly discussing hurdles to B. hilaris SIT. Is mass-rearing possible? Have release methods been developed? How much damage do males cause relative to females? Such a discussion would provide the reader with a broader view of issues and applicability of SIT.

The English is very good and the statistical treatment of results and their description is clear. I think it might help to emphasize more clearly in the Results or Discussion what radiation dose would be most effective in a release program based on some simple models. Nothing elaborate, particularly since male mating competitiveness is not known.

A scatter plot of egg number vs. hatching rate might be an effective way to visually represent this.

I did not understand the idea about promoting parasitoids that is mentioned in the Introduction and Discussion. Please expand on this further.

I would like to see the authors spend a bit more time in the Introduction on the factors that might affect fecundity in a polyandrous species as an effect of SIT. The authors emphasize the effect that they manipulated: sterility conferred by sperm. But likelihood of mating and sperm number of irradiated males relative to controls is also important.

The description of the mating order experiments becomes tedious after a while. I suggest that the authors consider some kind of simplifying abbreviation such as I-N, N-N, N-I or similar.

Reviewer 3 Report

The document is well-written, clear, timely, and easy to follow. The introduction clearly describes the subject, and the materials, and methods are complete and easy to understand. The results are written in a complete way. The discussion is appropriate.

Only, it is recommended that the authors add at the end of the discussion, according to their results, what impact would the frequent release of sterile insects has, what would be the release frequency, and the possible sterile: fertile ratio. And in the conclusion say which dose is the one that the authors recommend to use.

Round 2

Reviewer 1 Report

Significant changes are required to make this manuscript publishable, especially setting clear objectives, methodologies aligned with objectives, precise result presentations, elaboration of research findings in light of past corresponding studies.

It is evidenced that English level should be upgraded, making it simple and understandable.